# Toxicity and Risks Assessment of Polycyclic Aromatic Hydrocarbons in River Bed Sediments of an Artisanal Crude Oil Refining Area in the Niger Delta, Nigeria

Ibigoni C. Howard [1,2] , Kingsley E. Okpara [1] and Kuaanan Techato [1,*]

1   Faculty of Environmental Management, Prince of Songkla University, Hat Yai 90110, Songkla, Thailand; dromiete_ib@yahoo.com (I.C.H.); okparakingsley777@gmail.com (K.E.O.)
2   Department of Chemistry/Biochemistry, Federal Polytechnic, Nekede, Owerri 460113, Nigeria
*   Correspondence: kuaanan.t@psu.ac.th

**Abstract:** Polycyclic aromatic hydrocarbons (PAHs) are ubiquitous environmental pollutants that possess serious risks to human health and the environment. Forty riverbed sediments samples were collected in mangrove river bed sediments where artisanal refining of crude oil takes place in the Niger Delta of Nigeria. The concentration, occurrence, distribution, toxicity and health risk of sixteen priority PAHs (16PAHs) were analysed in the samples. Apart from Nap, Acy, BkF, InP and DbE, all the other PAHs were present in all the sampled points of the studied area with BbF and BaA recording the highest mean values. The range and mean of the total PAHs ($\sum$16PAHs) of this study are 23.461–89.886 mg/kg and 42.607 $\pm$ 14.30 mg/kg dry weight (dw), which is classified as heavily contaminated when compared to the European classification of PAHs pollution in soil (>1.0 mg/kg). The range of the effect range factors used to assess the risk of PAHs in an ecosystem (Effect rang-low (ER-L) and Effect range-median ER-M) of this study is from 0.953 to 8.80 mg/kg. PAHs below ER-L (4.0 mg/kg) indicate no toxic effect, but values above ER-M (44.79 mg/kg) indicate toxic effects to the sediments, its resources and, ultimately, the public that consumes the resources thereof; hence, the study area falls within the contaminated category. The occurrence of the high molecular weight (HMW) PAHs (73.4%) supersedes those of the lower molecular weight (LMW) PAHs (26.6%). The diagnostic ratios and principal component analysis suggest that the main contributors of PAHS into the sediments are the combustion of biomass, fossil fuel (crude oil) and pyrogenic sources. The toxic equivalent quotient (TEQ) and mutagenic equivalent quotient (MEQ) of PAHs ranged from 2.96 to 23.26 mgTEQ/kg dw and 4.47 to 23.52 mgMEQ/kg dw, and the total mean toxic equivalency quotient ($\sum$TEQ) (15.12 $\pm$ 8.4 mg/kg) is also greater than the safe level of 0.6 mg/kg, which indicates high toxicity potency. The mean incremental lifetime cancer risks (ILCRs) of human exposure to PAHs shows that both adults Total$_{ILCR\ adults}$ ($6.15 \times 10^{-5}$) and children Total$_{ILCR\ children}$ ($2.48 \times 10^{-4}$) can be affected by dermal contact rather than ingestion and inhalation. Based on these findings, the appropriate regulatory bodies and other organs of government in the region should enforce outright stoppage of the activities of these illegal artisans who do not have control mechanisms for loss control at the site and carry out appropriate clean-up of the area.

**Keywords:** polycyclic aromatic hydrocarbons; diagnostic ratios; risks; human exposure; toxicity

## 1. Introduction

The contamination of aquatic resources has become a daily occurrence especially in developing nations where immense wastes are generated without proper waste management practices and technology to treat such wastes effectively [1,2]. These wastes are basically a result of the increased rate and technology utilized in the extraction of economically valued resources such as petroleum, gold, etc. The processes of production, refining and marketing of these mineral resources result in the discharge of enormous wastes, either in gaseous, liquid and/or solid forms. Waste assimilation and transportation capabilities of

most environmental systems have simply been overstretched. This is as a result of the un-regulated and continuous discharge of urban, agricultural and industrial wastes into these systems. Most of these wastes generated include chemical toxicants such as Polyaromatic hydrocarbons (PAHs), Polychlorinated biphenyls (PCBs), heavy metals, etc., which persist in the general environment and are ultimately transported into the aquatic environment and, hence, bioaccumulate in aquatic resources of contaminated water bodies [3,4].

To add to this dilemma of improper waste management, Yabrade and Tance [5], Howard et al. [6] in their various studies concluded that the unwholesome use of stolen crude oil by balkinising crude oil facilities for pecuniary gains through the use of crude refining methods and discharge of petroleum products and crude oil into the environment with outright disregard for the safety of the environment and its host is another burden. This irresponsible act of national theft endangers not only the natural resource base of the area but also the livelihood of the natives who depend on the aquatic resources for their daily survival. It is, therefore, of immense importance to monitor, assess and safeguard the contamination of aquatic resources through indiscriminate discharges of wastes [5–7].

Polyaromatic hydrocarbons have been known to be persistent organic compounds (POCs) that are present in various media in the environment such as soil, water, sediments, air and aquatic organisms [4,8–11]. The main natural sources of PAHs are volcanic eruptions and wild bush fires [12]. On the other hand, some PAHs are byproducts of mankind's day-to-day operations such as PAHs released due to incomplete combustion of PAHs containing substances such as garbage, pesticides, wood, dyes tobacco, gas, oil, coal and plastics. Others include oil spills, urban runoff, domestic and industrial wastewater discharges, vehicle exhaust and industrial emissions [13,14].

Among the four categories of PAHs inputs to the environment (biogenic PAHs produced by living organisms; pyrogenic or pyrolytic PAHs are derived from incineration processes; petrogenic PAHs are those derived from fossil fuels; and diagenic PAHs are those derived from transformation process in soils and sediments). By using PAH concentrations or molecular ratios, several authors [8,15–17] have averred that it is only the pyrolytic (pyrogenic) and petrogenic PAHs that are more in abundance and generally formed by the reduction in biogenic precursors. For instance, Ambade et al. [4] in their study observed that "89% ratios of AN/(AN + Phen) were >0.1 and 11% of ratios were <1.0 which showed combustion of petroleum sources, 63% ratios of Flur/(Flur + Pye) were >0.5, 31% were <0.5 and 5% or ratios were equal to 0.5 which showed a source of pyrolytic origin and petrogenic origin, respectively." Adeniji et al. [18]. provided a value of $0.44 \pm 0.06$ for AN/(AN + Phen), and $0.4 \pm 0.09$ for Flur/(Flur + Pye), which is an indication of pyrolytic and petrogenic sources. These four categories produce different kinds of PAHs, and it is possible to identify the contribution from each to the total PAHs in an environmental sample.

PAHs are non-essential for the growth of plants, animals, or humans and are shown to pose potential deleterious effects on a range of aquatic organisms. They are listed in the category of carcinogenic and toxic contaminants found in aquatic ecosystems [4,17,18] and have adverse health effects; oral intake of PAHs through food, inhalation, and dermal interaction causes a significant risk to human health. They pose a serious threat to the health of aquatic organisms and human life through bioaccumulation within short periods of exposure [14,19–22]. They are non-ionic, highly hydrophobic and sometimes hydrophilic and readily adsorbed onto particulate matter and become deposited in the sediment. Thus, coastal and marine sediments become the ultimate sinks for PAHs where the degradation of PAHs is particularly slow [13,20,23]. By the European classification of PAHs pollution in soil, the total PAH level of <0.200 mg/kg is regarded as the allowable level in soil while that which is >1.0 mg/kg is regarded as heavily contaminated [24–26].

PAHs can further be categorized through their molecular weights: Low molecular weight (LMW) PAHs' that are made up of more than one fused aromatic ring of carbon and hydrogen atoms and the higher molecular weight (HMW) PAHs that are made up of four or more benzene rings and are most stable and persistent than the LMW PAHs. Currently, about 200 PAHs have been found in different environmental media [4]. Due

to their carcinogenic, mutagenic and toxic potentials, PAHs are of serious environmental concern; hence, the United States Environmental Protection Agency (USEPA) has declared sixteen PAHs as priority pollutants [27].

In recent times, several workers have reported PAHs values in different segments of the environment here in the Niger Delta area of Nigeria; for instance, Ref. [28] reported the concentration of PAHs in selected sediments of the Niger Delta region to be below the detectable limit at 1821.5 mg/kg. Agbozu et al. [29] also reported PAHs concentrations of 3850.9 to 7681.4 µg/g dry weights (dw) in suspended particulate matter (SPM) from an urbanized river system; they went further to assert by using principal component analysis that petroleum combustion is the major PAH source in SPM from the aquatic ecosystem and that the relative PAH patterns in SPM were found to correlate with the PAH patterns of different potential contamination sources. Other studies on PAHs exposure around the globe that have raised serious concern are those reported by [9] in urban soil in the typical semi-arid city of Xi'an in Northwest China wherein the total concentration of the sixteen PAHs ($\sum$16PAHs) in the urban soil ranged from 390.6 to 10,652.8 µg/kg with a mean of 2052.6 µg/kg, and the concentrations of some individual PAHs in the urban soil exceeded Dutch Target Values of Soil Quality, and the 16 PAHs represented heavy pollution. Adeniji et al. [18] reported that the total concentrations of the PAHs in water and sediment samples of Buffalo River Estuary, South Africa, varied at 14.91–206.0 µg/L and 1107.0–22,310.00 µg/kg, respectively, and these total levels of PAHs were above the target values in the two media; again, Dong et al. [30] reported PAH levels ($\sum$16PAHs) of 5736.2 to 69,362.8 ng/g in the sediments of River Taihu Lake, China.

The study area commonly known as the Oturuba River in the Andoni Local government area of Rivers State is a major mangrove wetland area in the Niger Delta of Nigeria that involves lots of fishing, transportation, tourism and farming activities. It also has a network of crude oil pipelines and petroleum facilities within the area. Recently, there is artisanal refining of crude oil in the area, which upon physical observation has greatly impacted the ecological system of the area; dried crude oil can be seen at the adjourning lands while crude oil sheen can be seen on the surface of the sediment, mangrove roots and water, which has resulted in stunted mangrove trees, retarded stem girth of mangrove trees and roots, low photosynthetic rates, etc.; hence, there is a need to assess the level and distribution patterns of PAHs ($\sum$16PAHs) in river bed sediments in order to provide information on the possible origins and possible risks to the environment and human health of the inhabitants of the area.

## 2. Materials and Methods

### 2.1. Description of the Study Area

The study area has been described in an earlier study [6]. Briefly, the study area is located at the southwest side of the Andoni local government area of Rivers State Nigeria between longitude 7°20′48.639″ E and latitude 4°27′30″ N where the makeshift crude oil refinery is. The river known as the Oturuba river is tidal and runs through interconnections of mangrove swamps/trees on both sides into the Andoni River (Okwaan Obolo), which is connected to the Atlantic Ocean. The area is flat and about 5 m above sea level; it is a replica of the Niger Delta mangrove terrain that is conspicuously sported with thick, soft and very soft mangrove soils also known as Chikoko (a mixture of acid sulphate, silty clay, clay loam and peat) with a hydrogen potential (pH) of 4 for newly deposited soft soil (mainly made up of silt) at low tides, pH of 6 in high tide locations and pH of 7 for transitional swamps at high tide [31]. Forty sampling stations ($S_1$–$S_{40}$) were selected within the study area (Figure 1).

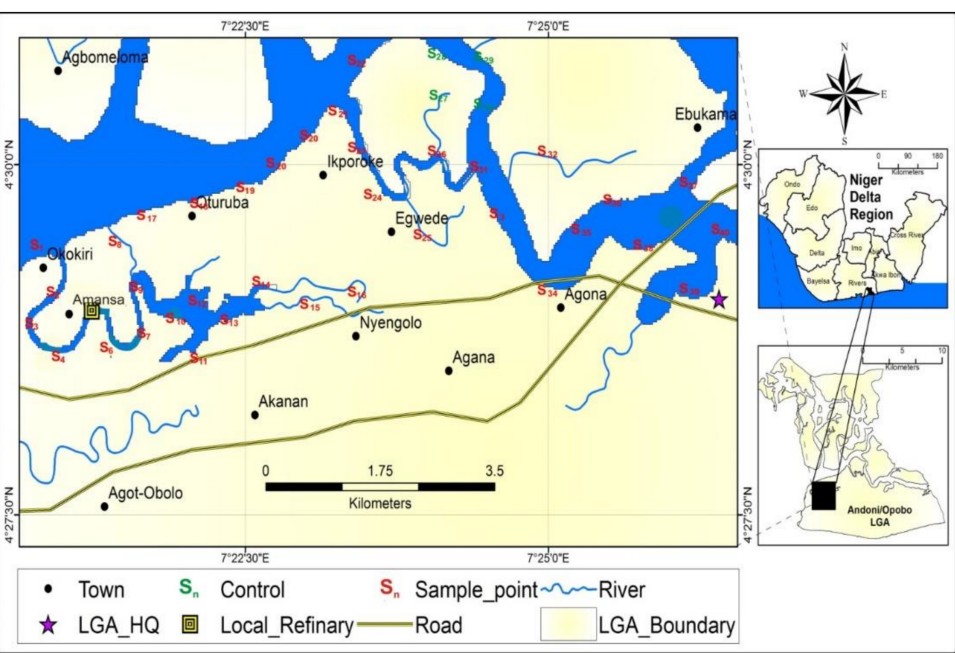

**Figure 1.** Map of the study area.

## 2.2. Sample Collection, Storage and Pretreatment

River bed sediment samples of about 250 g each were collected in triplicates at random with the aid of a stainless-steel van Veen grab sampler [4]. The samples were placed in plastic bags, properly labelled and stored in a sample chest; thereafter, they were transported to the laboratory on the same day. The semi-solid samples were kept in the refrigerator at 4 °C until analysis. Composite mixtures were made for each set of samples, and out of it a portion was weighed to obtain wet weight; then, they were air-dried in a dark location for five days, and the dry weight was obtained too. The dried samples were then macerated into a powdery form and sieved with a 0.5 mm sieve. The sieved samples were kept in well-labelled smaller sample containers with a lid, from which samples were withdrawn for analysis.

## 2.3. Extraction, Clean-Up and Analysis of PAHs from Samples

PAHs were extracted from the sediment samples and also cleaned up by using a modified form of the USEPA method [32,33], respectively. An amount of 5.0 g of dried sediment samples was weighed out and mixed with anhydrous sodium sulfate (1:5 ratio of sediment to sodium sulfate). The samples were extracted with Solid Phase Extractor (SPE) with n-hexane and dichloromethane (DCM) (1:1 $v/v$, 50, 25 and 25 mL). The combined supernatant was filtered through 5 g of silica gel column with 25 mL 1:1($v/v$) elution of hexane and DCM [27]. The evaporation of solvent fractions was carried out with the aid of a rotary evaporator (Buch R-3000) and acetonitrile was added as an "exchange solvent" and evaporated to a final volume of 2 mL. The extract was again passed through a silica gel glass column (30 cm long and 3 cm diameter glass) filled with 10 g activated silica gel and 5 g anhydrous $Na_2SO_4$. The PAHs were later eluted using 100 mL of DCM and n-hexane mixture (1:1 $v/v$) again. Finally, under a gentle stream of pure nitrogen, the eluent solvent was concentrated to about 1 mL. All sample extracts were kept in amber glass vials at $-4$ °C until they were analysed within the week of extraction.

## 2.4. Instrumental Analysis

The separation and identification of all samples were analyzed on an Agilent 6890N gas chromatograph equipped with a Flame Ionization Detector (FID). One micro litre of each of the extracted samples was injected in a splitless model into the GC-FID equipped with a column, which was DB-5 or HP-5 (30 m × 0.25–0.32 mm; ID × 0.25 μm film

thickness). Helium was used as carrier gas at a constant flow rate of 1 mL/min. Before injecting samples, the column temperature was programmed from an initial temperature of 50 °C (5 min hold) to 200 °C (5 min hold) at the rate 10 °C/min and then followed to 260 °C (15 min hold) at the rate 6 °C/min. FID was operated in the electron impact mode at 70 eV. The transfer line and ion trap manifolds were set at 280 °C and 230 °C, respectively. Quantification was performed by the external standard method by using a 16 PAHs standard mixture (AccuStandard PAH Mix, 1 mg/mL in methylene chloride). The 16 PAHs detected in sequence were as follows: naphthalene (Nap); acenaphthylene (Acy); acenaphthene (Ace), fluorene (Flu); phenanthrene (Phe); anthracene (Ant); fluoranthene (Flt); pyrene (Pyr); benz[a]anthracene (BaA); chrysene (Chry); benzo[b]fluoranthene (BbF); benzo[k]fluoranthene (BkF); benzo[a]pyrene (BaP); indeno[1,2,3-cd]pyrene (IndP); dibenz[a,h]anthracene (DahA); benzo[ghi]perylene (BghiP). The final quantification of each of the individual PAHs was later recalculated into a dry weight basis of mg/Kg by using the following Equation (1) [34]:

$$GC\ reading(\mathrm{mg/L}) \times \frac{Volume\ of\ Solvent(\mathrm{L})}{Weight\ of\ Sample(\mathrm{Kg})} \times \frac{100}{D} \tag{1}$$

where $D$ is the percentage dried sample, the volume of solvent used is 100.0 mL and the weight of the sample used is 5.0 g.

## 2.5. Quality Assurance and Quality Control

For quality assurance and quality control, replicate samples (field blanks) were analysed for all samples, and reagent blanks were also analysed after every five samples [26]. The reagent blank containing the surrogate standard (AccuStandard o-Terphenyl) and the solvent was analysed in order to evaluate the interference and contamination of the solvents, reagents and glassware used. Again, the processes of extraction, clean-ups and setting up the instrumental system were examined by spiking each real sample and reagent blank with the surrogate internal standard (o-terphenyl) of a known concentration. The recovery percentages of 16 PAHs in the samples were approximately 93.0–100.0%. The $r^2$ values for the calibration ranged from 0.992 to 0.9999; the method detection limits (MDLs) of the target PAHs were calculated as three times the standard deviation (SD) plus the mean concentrations of target compounds in blank samples. The MDL of the individual PAHs ranged from 0.022 to 0.120 μg/L. The concentration of PAHs less than the method detection limits was deemed as non-detected (ND). The sources of PAHS in the samples were analysed by using the different isomeric ratios and principal component analysis.

## 2.6. Risk Assessment of PAHs

Due to hydrodynamics of any river system, pollutants accumulated in river bed sediments become re-suspended into the water column and, thus, pose a potential ecological risk to not only aquatic resources but also to the inhabitants that subsist on such resources as is the case in most developing countries including a typical Niger Delta terrain. [1,5,6,25,35,36]. The risk of exposure to PAHs in the river bed sediments in the study area was evaluated by using the benzo(a)pyrene toxic equivalency quotient (TEQ); effect-based guidelines–effect range median (ER-M); and effect range-low (ER-L) and incremental lifetime cancer risk (ILCR) factors. The risks of PAHs in sediment/and other environmental media have been studied in the literature by using BaP carcinogenic (BaP$_{TEQ}$) and BaP mutagenic (BaP$_{MEQ}$) equivalency factors [4,9,14,18,25,27,37].

The BaP carcinogenic equivalency factor (BaP$_{TEQ}$) for the PAHs was evaluated using the following equation:

$$\mathrm{BaP_{TEQ}} = \sum \mathrm{C_i} \times \mathrm{BaP_{TEF}} \tag{2}$$

where BaP$_{TEF}$ is the cancer potency relative to BaP, and C$_i$ is the individual PAH concentration.

The BaP mutagenic equivalency factor ($BaP_{MEQ}$) for the PAHs was evaluated by using the following Equation:

$$BaP_{MEQ} = \sum C_i \times BaP_{MEF} \tag{3}$$

where $BaP_{MEF}$ is the mutagenic potency relative to BaP, and $C_i$ is the individual PAH concentration.

The BaP carcinogenic equivalency factors ($BaP_{TEFs}$) of the seven carcinogenic PAHs used were BaP (1), BaA (0.1), BbF (0.1), BkF (0.01), Chry (0.001), DahA (1) and IndP (0.1) [37]. The BaP mutagenic potency factors (BaPMEFs) were BaP (1), BaA (0.082), BbF (0.25), BkF (0.11), Chry (0.017), DahA (0.29) and IndP (0.31) [14,18,25,37].

The toxicity of PAHs has also been assessed in sediments and other environmental media by using effect-based guidelines–effect range median (ER-M) and effect range low (ER-L) developed by environmental professionals [4,9,14,25]. ER-M and ER-L values are measures of toxicity in an environmental medium (e.g., marine sediments) used in formulating guidelines in assessing toxicity hazards of contaminants. For instance, the normal level of total PAH in sediments should be below the effect range-low (4.0 mg/kg), while abnormal levels are regarded as those above the effect range-median (44.792 mg/kg) [4,9,12,14,18,37,38]. This abnormal level indicates toxic effects relative to the medium (sediments), its resources and ultimately the public that consumes the resources thereof. In this study, the mean values of each of the 16PAHs were compared with the established ER-M and ER-L values.

## 2.7. Incremental Lifetime Cancer Risk (ILCR)

By the exposure of the locals who subsists on the aquatic resources of the area daily, there is a need to assess the risk of human exposure to PAHs in the river bed sediments, and this was carried out by using the ILCR model of USEPA, which considered three major routes of human exposure to contaminants (i.e., ingestion, dermal and inhalation of vapor or sediment/dust [39]). ILCR is used to quantify the carcinogenic risk to human beings when exposed to PAHs in an environment [9]. The total carcinogenic risk was calculated as the summation of the individual risks from the three exposure routes. The parameters and model equations adopted for the assessment of ILCR are shown in Table 1 and Equations (4)–(7), respectively [39–41]:

$$ILCR_{ing} = \frac{(C_{sed} \times IngR \times EF \times ED \times CF \times SFO)}{BW \times AT} \tag{4}$$

where $ILCR_{ing}$ is the incremental lifetime cancer risk via ingestion of sediment particles; $C_{sed}$ is the concentration of the pollutant in the sediment (mg/kg); IngR is the ingestion rate of sediment (mg/day); EF is the exposure frequency (days/year); ED is the exposure duration (years); BW is the average body weight (kg); AT is the averaging time (days); CF is the conversion factor ($1 \times 10^{-6}$ kg/mg); and SFO is the oral slope factor $(mg/kg/day)^{-1}$. The SFO values $(mg^{-1} kg^{-1} day^{-1})^{-1}$ were BaA = $7.3 \times 10^{-1}$; Chry = $7.3 \times 10^{-3}$; BbF = $7.3 \times 10^{-1}$; BkF = $7.3 \times 10^{-2}$; BaP = 7.3; IndP = $7.3 \times 10^{-1}$; and DahA = 7.3 [39]:

$$ILCR_{derm} = \frac{(C_{sed} \times SA \times AF_{sed} \times ABS \times EF \times ED \times CF \times SFO \times GIABS)}{BW \times AT} \tag{5}$$

where $ILCR_{derm}$ is the incremental lifetime cancer risk via dermal contact of sediment particles; SA is the surface area of the skin that is in contact with sediment ($cm^2$/day); AF is the skin adherence factor for dust/sediment (mg/$cm^2$); and ABS is the dermal absorption factor (chemical specific).

$$ILCR_{inh} = \frac{(C_{sed} \times EF \times ET \times ED \times IUR)}{PEF \times AT^*} \tag{6}$$

**Table 1.** Parameters for estimating human cancer risk.

| Exposure Variables | Age | | Source |
|---|---|---|---|
| | Child | Adult | |
| Body weight, BW (kg) | 15 | 60 | [42] |
| Exposure duration, ED (years) | 6 | 24 | [43] |
| Exposure frequency, EF (days/year) | 313 | 313 | [44] |
| Averaging time, AT (days) | $52 \times 365 = 18{,}980$ | $52 \times 365 = 18{,}980$ | [45] |
| Ingestion rate, IngR (mg/kg) | 200 | 100 | [43] |
| Adherence factor, AF (mg/cm$^2$) | 0.2 | 0.07 | [43] |
| Adsorption fraction, ABS (unitless) | 0.13 | 0.13 | [43] |
| Particle emission factor, PEF (mg$^3$/kg) | $1.36 \times 10^9$ | $1.36 \times 10^9$ | [43] |
| Exposure skin area, SA (cm$^2$) | 2800 | 5700 | [45] |
| Averaging time, AT_ (h) | 52 yrs $\times$ 365 days/yr $\times$ 24 h/day = 455,520 | 52 yrs $\times$ 365 days/yr $\times$ 24 h/day = 455,520 | [43] |
| Gastrointestinal absorption factor, GIABS | 1 | 1 | [46] |
| Conversion factor, CF | $1 \times 10^{-6}$ | $1 \times 10^{-6}$ | [44] |
| Exposure time, ET (hr/day) | 8 | 8 | [43] |

The average life expectancy of Nigeria is 52 years [25]. GIABS is the gastrointestinal absorption factor.

In the above equation, $ILCR_{inh}$ is the incremental lifetime cancer risk via inhalation of sediment particles; ET is the exposure time (h/day); and IUR is the inhalation unit risk $(mg/m^3)^{-1}$. The IUR values $(\mu g\ m^{-3})^{-1}$ were $1.1 \times 10^{-4}$ for BaA, BaF, BkF and IndP while $1.1 \times 10^{-5}$ for Chry, $1.1 \times 10^{-3}$ for BaP and $1.2 \times 10^{-3}$ for DahA [46]. AT* is the averaging time (h). and PEF is the particle emission factor = $1.36 \times 109\ m^3/kg$. PEF is the inhalation of pollutants that were adsorbed by respirable particles ($PM_{10}$) and relates to the concentration of a pollutant in soil with the concentration of respirable particles in the air due to fugitive dust emissions [25].

Total Carcinogenic Risk

The summation of the three different forms of risk provides the total carcinogenic risk [12,14,41].

$$\text{Total ILCR} = \text{ILCRing} + \text{ILCRinh} + \text{ILCRdrem} \tag{7}$$

The New York State Department of Health regulatory guideline [41] indicates that $ILCR \leq 10^{-6}$ values show no risk or negligible risks that are similar to those obtained from exposure to X-ray, while $\geq 10^{-4}$ values show high risk with adverse health effects such as cancer. They provided general classification $total_{ILCR}$ as follows: $\leq 10^{-6} \cong$ very low; $10^{-6}$ to $10^{-4} \cong$ low; $>10^{-4}$ to $10^{-3} \cong$ moderate; $>10^{-3}$ to $10^{-1} \cong$ high; and $\geq 10^{-1} \cong$ very high [41].

## 3. Results and Discussion

The result of polycyclic aromatic hydrocarbons recorded in the study area is summarised in Figure 2, which shows the least observed value of 38.1 mg/kg for Nap and the highest value of 352.1 mg/kg for BbF. The individual mean range of the study is 23.4–89.9 mg/kg, and the total mean ($\sum$16PAHs) is 42.6 $\pm$ 14.3 mg/kg, as shown in Table 2. The most abundant PAHs of this study from Figure 2 are BbF (20.6%) and BaA (20.7%), followed by BaP (9.1%). Apart from Nap, Acy, BkF, InP and DbE, all the other PAHs were

completely detected in all forty sampling points (note: % ring PAH = Sum of each PAH Concentration with the same ring/Total concentration of all PAHs × 100).

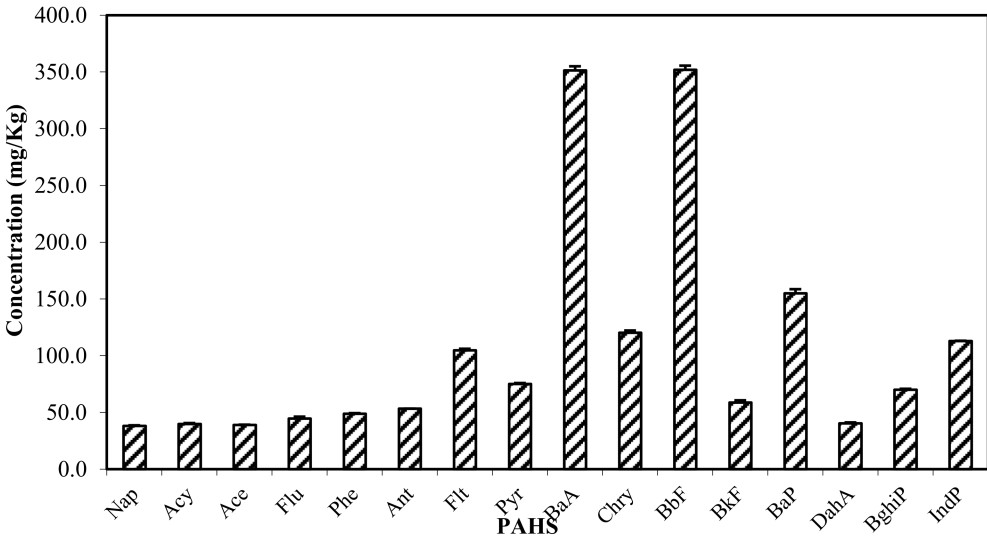

**Figure 2.** Individual PAHs levels in the sediments of the study area.

**Table 2.** Descriptive statistics, ER-L and ER-M values of PAHs contents in the sediment of the area studied (in mg/Kg).

| PAHs | Min | Max | Mean ± SD | %PAHs | ER-L | ER-M | Mean Values of This Study |
|------|-----|-----|-----------|-------|------|------|---------------------------|
| Nap | ND | 3.377 | 0.953 ± 0.61 | 2.27 | 0.160 | 2.100 | 0.953 * |
| Acy | ND | 3.851 | 0.995 ± 0.28 | 2.33 | 0.044 | 0.640 | 0.995 ** |
| Ace | 0.462 | 3.062 | 0.976 ± 0.63 | 2.27 | 0.016 | 0.500 | 0.976 ** |
| Flu | 0.420 | 11.558 | 1.113 ± 1.71 | 2.61 | 0.019 | 0.540 | 1.113 ** |
| Phe | 0.546 | 2.520 | 1.226 ± 0.40 | 2.88 | 0.240 | 1.500 | 1.226 * |
| Ant | 0.588 | 2.016 | 1.327 ± 0.40 | 3.12 | 0.085 | 1.100 | 1.327 ** |
| Flt | 0.378 | 5.880 | 2.624 ± 0.33 | 6.16 | 0.600 | 5.100 | 2.624 * |
| Pyr | 0.630 | 3.150 | 1.882 ± 1.23 | 4.42 | 0.665 | 2.600 | 1.882 * |
| BaA | 2.835 | 16.367 | 8.790 ± 3.33 | 20.63 | 0.261 | 1.600 | 3.008 ** |
| Chry | 1.008 | 8.400 | 3.008 ± 1.97 | 7.06 | 0.384 | 2.800 | 8.790 ** |
| BbF | 0.895 | 16.477 | 8.803 ± 3.46 | 20.67 | 0.320 | 1.800 | 8.803 ** |
| BkF | ND | 10.769 | 1.472 ± 1.91 | 3.46 | 0.280 | 1.620 | 1.472 * |
| BaP | 1.176 | 14.822 | 3.878 ± 3.42 | 9.10 | 0.430 | 1.600 | 3.878 ** |
| InP | ND | 4.990 | 1.005 ± 1.05 | 2.36 | 0.240 | - | 1.005 ** |
| DahA | ND | 4.637 | 1.747 ± 0.92 | 4.10 | 0.063 | 0.260 | 1.747 ** |
| BghiP | 1.344 | 3.990 | 2.817 ± 0.56 | 6.61 | 0.085 | 1.600 | 2.817 ** |
| ∑16PAHs | 23.461 | 89.886 | 42.607 ± 14.30 | | **—higher than both ER-L and ER-M; *—Higher than ER-L | | |
| ∑7CPAHs | 16.187 | 64.479 | 29.773 ± 10.31 | | | | |
| ∑COMBPAH | 11.319 | 41.933 | 17.428 ± 6.52 | | | | |
| LMWPAHs | 4.221 | 20.837 | 6.581 ± 3.29 | | | | |
| HMWPAHs | 18.875 | 77.029 | 36.02 ± 11.95 | | | | |

∑16PAHs: the sum of sixteen individual PAHs; COMB PAH: the sum of major combustion-specific compounds containing Fla, Pyr, BbF, BaP, BkF, Chy, BghiP, BaA and InP; ∑7CPAHs: the total of seven carcinogenic PAHs including BkF, Chy, BbF, DahA, BaP, BaA and InP; ∑LMWPAHs: the total of low molecular weight PAHs, i.e., NaP, Ace, Acy, Phe, Flu and Ant; HMWPAHs: the sum of high molecular weight PAHs, i.e., Fla, Pyr, BaA, Chy, BkF, BbF, DahA, BaP, BghiP and InP; SD: standard deviation; ND: not detected; ER-M—effect range Median; ER-L—effect range Low.

The sixth and fifth ring PAHs (36.95% and 34.36%) were more dominant in the sediment samples of the study area than all others; however, the three-ring PAHs had 16.18% while that of the four-ring PAHs is 10.35%, and the two ring PAHs is 2.16 %, which indicates that the higher molecular weights (HMW) PAHs are more abundant than the lower

molecular weight (LMW) PAHs in the ratio of 5.54:1.0 (36.0:6.5 mg/kg). This attribute of molecular weight of PAHs (high or low) has been used in the literature to ascertain the possible sources of PAHs in any medium [14,18,25]. Again, the toxic levels of PAHs such as BbF, BaA, BaP and Chyr were found to be greater than the other toxic PAHs with mean values of 8.8 ± 199.1, 8.8 ± 198.7, 3.9 ± 88.1 and 3.0 ± 67.7 mg/kg, respectively.

The total mean range of PAHs (∑PAHs) levels of this study, 23.5–89.9 mg/kg, is within the range of related studies in the Niger Delta of Nigeria but above most values from similar studies around the world. For instance, the range of this study is far below that obtained by [47] (Table 3) in the Woji Creek, which is an urbanized area where the industries in the Trans Amadi Industrial layout of Port Harcourt discharges their effluent into. Again, it is still below that obtained by [28] who reported PAH concentrations of 3850.9 to 7681.4 µg/g dry weights in suspended particulate matter (SPM) from an urbanized river system. However, the range is below that reported by [4] in the sediments of the Damodar Basin in India [48], in the sediments of the White Nile, East Africa, and that of [29] in the sediments of Taihu lake in China. Table 3 shows a comparison of the concentration of PAHs in the sediments of this study and other related studies around the world.

**Table 3.** Comparison of PAHs levels in sediments of this study with related studies (mg/kg).

| Study Area | No. of PAHs Studied | PAHs Range and/or (Mean) | References |
|---|---|---|---|
| Warri River at Ubeji, Nigeria | 16 | (4.588) | [49] |
| Abeokuta Metropolis, SW, Nigeria | 16 | 11.9–41.6 | [50] |
| Onitsha Nigeria | 13 | 0.01–4.281 | [51] |
| Woji Creek, in the Niger Delta | 16 | 687.93–1821.5 | [47] |
| River Ethiope, Delta State, Nigeria | 16 | 0.185–3.679 | [52] |
| Niger Delta Region of Nigeria | 16 | BDL to 1821.5 | [28] |
| Middle region of Huaihe River. Chian | 16 | 0.072–0.139 | [53] |
| Isfahan, Iran | 16 | 0.058–11.730 (2.001) | [54] |
| Hoogly and Brahmaputra river, India | 16 | 0.00–0.636 | [55] |
| Semi-arid city of Xi'an in Northwest China | 16 | 0.391–10.652 (2.053) | [9] |
| Buffalo River Estuary South Africa | 16 | 1.107–22.310 | [18] |
| River of Taihu Lake, China | 16 | 5.736–69.363 | [30] |
| South Part of Al-HammerMarsh, Southern Iraq | 16 | 0.0428–0.434 | [56] |
| White Nile, East Africa | 16 | 0.566–0.674 | [48] |
| Damodar basin, India | 16 | 0.00–0.582 | [4] |
| Oturuba River Niger Delta | 16 | 23.461–89.886 (42.607 ± 14.30) | This study |

### 3.1. PAHS Composition

The sixteen (16) priority PAHs have been classified into low (2 + 3-ring), middle (4-ring) and higher (5 + 6-ring) molecular weight PAHs. Figure 3 is a compositional diagram (a pie chart that compares multiple factors at a time) that shows the degree of the composition of each of the classes in the sediments of the study area. It shows that the HMW PAHs (73.4%) are more abundant than the middle and low MWPAHs (26.6%). This scenario could be due to HMW PAHs adsorbing to the sediment particles as earlier observed by [57–60] and corroborated by [4,25] in their various studies. Again, LMW PAHs are more volatile and more biodegradable, unlike the HMWPAHs that are more persistent in the environment [18]. Another possible reason for this is due to the non-bioavailability and less-water solubility properties of the HMW PAHs and high resistance to microbial degradation [61–63]. Han et al. [62] in their work opined that the higher concentrations of high molecular weight PAHs than all the other compounds in the total PAH concentration suggest that they can be attributed to the combustion of petroleum.

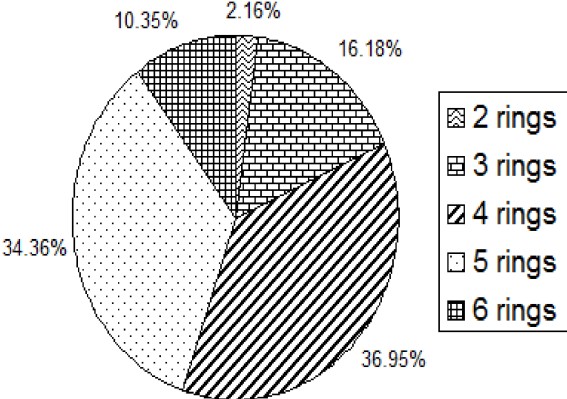

**Figure 3.** Pie chart diagram for the ring wise composition of different PAHs in the sediment samples of the study area.

### 3.2. PAHs Source Identification

Controlling the sources of PAHs and the quantitative comprehension of the various origins are pertinent for the evaluation and remediation of contaminated environments and health risks caused by these PAHs [64]. Natural and man-made sources are the basic sources of these pollutants in the environment. Out of these sources, the later possesses more danger to the environment and human health as their inputs into the environment are greater than natural sources [25]. The man-made sources of PAHs are classified as pyrolytic (pyrogenic) and petrogenic; the former includes the combustion of coal and petroleum (fossil fuel) and biomass while the latter is basically from crude oil and petroleum products.

The sources of selected PAHs have been identified by using concentration ratios or molecular ratios such as Flt/(Flt + Pyr), Ant/(Ant + Phe), Phe/Ant, BaA/(BaA + Chry), In/(In + BghiP) and low molecular weight/high molecular weight (LMW/HMW) [4,25,65–68]. For this study, AN/(AN + Phen) vs. Flur/(Flur + Pye) and IN/(IN + BgP) vs. BaA/(BaA + Chry) and LMW/HMW ratios were used as shown in Table 4 to determine the sources of PAHs in the sediments of the study area.

**Table 4.** Guidelines for PAHs diagnostic ratios considered for identifying sources of PAHs in the sediment samples of the study area.

| PAHs Molecular Ratios | Diagnostic Ratio | Sources | Reference | Range for This Study |
|---|---|---|---|---|
| AN/(AN + Phen) | <0.1 | Petroleum | [68] | 0.3–0.8 |
| | >0.1 | Combustion | | |
| Flur/(Flur + Pye) | <0.4 | Petrogenic | [69] | 0.2–0.8 |
| | 0.4–0.5 | Fuel combustion | | |
| | >0.5 | Coal, grass and wood burning | | |
| BaA/(BaA + Chry) | <0.2 | Petrogenic | [66] | 0.5–0.9 |
| | 0.2–0.35 | Fuel combustion | | |
| | >0.35 | Coal, grass and wood burning | | |
| In/(In + BgP) | <0.2 | Petrogenic | [67] | 0.3–1.0 |
| | 0.2–0.5 | Fuel combustion (crude oil or vehicular emission) | | |
| | >0.5 | Coal, grass and wood-burning | | |
| LMW/HMW | >1.0 | Petrogenic | [18] | 0.2–0.3 |
| | <1.0 | Pyrogenic | | |

The scatter plot in Figure 4a below indicates that 100% ratios of AN/(AN + Phen) were >0.1, which indicates that the combustion origin as the range of AN/(AN + Phen) was 0.3–0.8. Based on Table 4 and the scatter plot of Figure 4a, the range of the ratios of Flur/(Flur + Pye) is 0.2–0.8, which indicates petroleum and coal, grass and wood combustion sources. The range of In/(In + BgP), which is 0.3–1.0, indicates 100% mainly

combustion sources while that of BaA/(BaA + Chry) ranges from 0.5 to 0.9 of which 95% of the total value suggests pyrogenic and coal, grass and wood combustion sources, while 5% is of mixed sources, as shown in Figure 4b. The ratio of ∑LMW/HMW is lower than one in all sampling points (Table 4), which confirms the earlier assertion that HMW PAHs are more abundant than LMW PAHs in the sediments, implying dominant pyrogenic and combustion sources instead of petrogenic sources in the sediments [18].

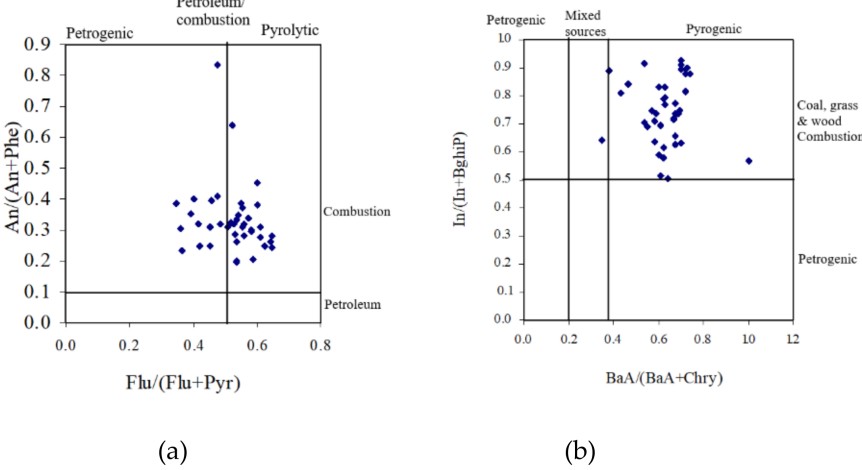

(a)                                            (b)

**Figure 4.** (**a**) 100% ratios of AN/(AN + Phen) were >0.1; (**b**) 100% mainly combustion sources while that of BaA/(BaA + Chry) ranges from 0.5 to 0.9.

*3.3. Principal Component Analysis*

Principal component analysis (PCA) is a statistical tool used to reduce high-dimensional data by bringing out the similarities, differences, trends and patterns into simpler dimensional sets of data known as principal components or extracts. The extracts are derived in decreasing order of importance so that the first principal component accounts for, as much as possible, the variation in the original data. For a set of data to be suitable for PCA, it must fulfill the KMO (Kaiser–Meyer–Oklin) sampling adequacy measures (>0.6–1.0) and Bartlett's test of sphericity [70,71]. In this study, the KMO sampling adequacy measure is 0.72, and Bartlett's test of sphericity has a Chi square of 476.151 and a *p*-value < 0.001. Again the sum of the principal components, which are also known as summary indices (PC- PC-1, PC-2, . . . , PC-n) from a given PCA, is always 100%, out of which the most significant components (PC1, PC2, etc.) which are indicated by the leveling off of the "Scree plot" (eigenvalues versus the component numbers) are extracted in order to explain the variations that exist in the original data set. From the analysis made using varimax rotation in SPSS (Statistical Package for Social Sciences) version 20, the most significant extracts are PC-1 (42.4%), PC-2 (16.3%) and PC-3 (9.7%) as they accounted for about 68.4% of the total variance (Table 5).

Principal component-1 (PC-1) in Table 5 explained 42.4% of the total variance and loaded with more LMW PAHs (Nap, Ace, Acy and Phe) and HMW PAHs (BkF, DahA, BaA, BaP and BhigP) and was less loaded with PAHs (Flur, Pyr, Chry, IndP, BbF, Ant and Flt). This indicates that these sets of PAHS are from the same sources as we have earlier observed in the diagnostic ratios. This finding agrees with that of Ambade et al. [4], who also observed the presence of (Nap, Ace and Acy) and (BkF, DahA and BhigP) in their first PCA extract (PC1) and Iwegbue and Obi [25] for LMW PAH ((Nap, Ace and Acy) in PC-1). The LMW PAHs are products of low-temperature pyrogenic processes such as biomass combustion [25]. PAH Nap is the main component of diesel fuels and gasoline, which may be formed by incomplete fuel combustion [4]. BkF is a product of wood combustion [72], while IndP is a product of pyrolysis or incomplete fuel combustion [73], and BgP is a marker of automobile emissions [65]. Thus, PC-1 describes PAHs sources due to incomplete fuel combustions and wood burning, which is actually occurring in the area due to the fact

that the artisans that refine crude oil in a crude manner use wood as their heating sources without regard to the safety of the environment.

**Table 5.** Extracted principal components PC-1, PC-2 and PC-3 analyses of the sediment samples.

| PAHs | PC-1 (42.4%) | PC-2 (16.3%) | PC-3 (9.7%) |
|---|---|---|---|
| Nap | 0.796 | −0.178 | 0.452 |
| Acy | 0.722 | −0.407 | 0.392 |
| Ace | 0.818 | −0.272 | 0.306 |
| Flu | 0.400 | 0.175 | 0.461 |
| Phe | 0.741 | −0.209 | −0.335 |
| Ant | 0.609 | 0.314 | |
| Flt | 0.615 | 0.541 | 0.142 |
| Pyr | 0.339 | 0.503 | −0.271 |
| BaA | 0.725 | 0.321 | 0.188 |
| Chry | 0.148 | 0.544 | −0.141 |
| BbF | 0.537 | 0.662 | −0.282 |
| BkF | 0.793 | −0.238 | −0.479 |
| BaP | 0.660 | −0.622 | |
| DahA | 0.837 | | −0.326 |
| BghiP | 0.749 | −0.251 | −0.341 |
| IndP | 0.465 | 0.514 | 0.297 |

Principal component-2 (PC-2) analysis explained 16.3%, while that of principal component-3 (PC-3) analysis explained only 9.7% of the total variance. PC-2 is predominantly loaded with PAHs such as BaP, BbF, Chry, IndP, Flt and Pyr. Both PC-1 and PC-2 are heavily loaded with BbF, BaP, Flt and IndP, which suggests that these PAHs (BbF, BaP, Flt and IndP) are from mixed sources [71]. For PC-3, the most heavily loaded PAH is BkF and Flu. It can be observed in Table 5 that in PC-2 and PC-3, there are negative loadings such as Ace, Acy, BkF, etc., for PC-2 and BkF, DahA, BGhP, etc., for PC-3; this implies a negative association or correlation of these PAHs with other PAHs where they originated from. The study area, which is known for bunkering and artisanal refining of crude oil into petroleum products without regards to the safety of the environment, portrays the result, herein presented (diagnostic ratios and PCA), to be mostly of pyrogenic origins and a consequence of the combustion of crude oil, wood and other biomass which they use to heat crude oil in metal containers. However, the abundance of HMW PAHs over LMW PAHs in sediments/dust has also been reported in related studies in the Niger Delta [25,48,53,74] and other regions of the world [4,9,14,26].

*3.4. Potential Ecosystem Risk Assessment*

Contamination levels of PAHs has been classified into four classes on the basis of total PAH ($\sum$PAH) concentrations—not contaminated (<0.200 mg/kg), slightly contaminated (0.200 to 0.600 mg/kg), contaminated (0.600 to 1.000 mg/kg) and heavily contaminated (>1.000 mg/kg) [24,25]. From these criteria, all the sampling points of this study fall within the contaminated to heavily contaminated classes. Moreover, the mean value of this study (42.607 ± 14.3 mg/kg) is greater than the heavily contaminated class; hence, the studied area can be classified as heavily contaminated. The range and mean of the study have already been compared with related studies around the world in Table 3. The value of total combustion-specific PAHs—Fla, Pyr, BbF, BaP, BkF, Chy, BghiP, BaA and InP—from the results ($\sum$COMBPAHs—17.428 ± 6.5) (Table 2) also explains this contamination level of the study area. This portends danger not only to the ecosystem but also to the inhabitants of the area that subsists on the aquatic resources of the area. The entire life of the indigenous people of the Niger Delta depends mostly on the aquatic resources of the area [1,4,5].

The effect-based guideline values, which estimate the relationship between a contaminant concentration and toxic response shown by an organism, were utilised to evaluate the potential toxicity levels of PAHs in the sediments [4,9,26,27]. The guideline values

were classified as effect range-low (ER-L) and effect range-median (ER-M) (Table 2). The total concentration of PAHs or each PAH above ER-M value (s) is an indication of the adverse effects on the ecosystem and likely danger to human health as well [4,9,14,18]. As presented in Table 2, the values of all the PAHs recorded in this study are all higher than the effect range-low values; Nap, Phe, Flt, Pyr and BkF were lower than the ER-M values while all the rest were 94.3% higher in value. The total mean of the sixteen PAHs (42.607 ± 14.3 mg/kg) is also above the ER-L (4.00 mg/kg) but slightly lower than the ER-M (44.792 mg/kg), and they are also above those reported by [4,27]. These high values of PHAs in the sediment of the study area imply that the aquatic resources of the area can be adversely impacted, although there could be some kind of uncertainties to this assertion as all natural and life sciences are subject to uncertainties [14,74]; hence, more research needs to be carried out, such as the levels of PAHs in water, biota (benthic organisms), mangrove roots, etc., with the sediments and the level of uncertainty factored into it.

### 3.5. Human Health Risk Assessment

Potential human health risks due to exposure of sediment-bound PAHs were evaluated by using Benzo(a)pyrene toxic equivalency quotient (BaP$_{TEQ}$), Mutagenic equivalency quotient (BaP$_{MEQ}$) and incremental life cancer risk (ILCR) methods. BaP$_{TEQ}$ is related to carcinogenicity, whereas BaP$_{MEQ}$ is the mutagenic effect, which includes noncancerous adverse effects such as impotency, low intelligent quotient birth defects pulmonary diseases, etc. [25]. The estimation of BaP$_{TEQ}$ and BaP$_{MEQ}$ for the study is shown in Table 6. The concentration range for BaP$_{TEQ}$ is 0.120–155.139 mg/kg, while that of BaP$_{MEQ}$ is 2.045–155.139 mg/kg. However, for both methods of risk assessment (BaP$_{TEC}$ and BaP$_{MEQ}$), BaP contributes more to carcinogenic and mutagenic potencies, and this is in line with the findings of [4,9,14,25]. The order of significant contributors to BaP$_{TEC}$ values is BaP > DahA > BbF > BaA, while that of BaP$_{MEQ}$ is BaP > BbF > IndP > BaA.

**Table 6.** Benzo(a)pyrene toxic equivalency quantities (BaP$_{TEQ}$) and mutagenic equivalency quantities (BaP$_{MEQ}$) levels.

| Assessment Method | BaA | Chry | BbF | BkF | BaP | DahA | IndP | Total | Mean ± Stdev |
|---|---|---|---|---|---|---|---|---|---|
| **BaP$_{TEC}$** | 35.161 | 0.120 | 35.210 | 0.589 | 155.139 | 40.207 | 11.269 | 277.695 | 6.94 ± 4.36 |
| **BaP$_{MEQ}$** | 28.832 | 2.045 | 88.026 | 6.477 | 155.139 | 11.66 | 34.933 | 327.112 | 8.18 ± 4.05 |
| | | | | | | | | 604.807 | 15.12 ± 8.41 |

Benzo(a)pyrene contains five rings, and this arrangement gives it a bay region often correlated with carcinogenic properties. The World Health Organisation recommended a guideline value of 0.7 μg/L [75] in drinking water, which corresponds to an excess lifetime cancer risk of $10^{-6}$, whereas the Netherlands National Institute of Public Health and the Environment's maximum permissible concentrations for *BaP* is 0.5 μg/L in water and 0.26 μg/g in soil [62]; hence, the high mean value of BaP (3.878 ± 3.42 mg/kg) in this study and other related studies reporting sediments/dust of the Niger Delta region at 1.15 μg/kg [76], 0.036 mg/kg [25] and 1.342 mg/kg [77] raises concern. The health concern of *BaP*, similarly to all other PAHs, is that its metabolic transformation by aquatic and terrestrial organisms into carcinogenic, teratogenic and mutagenic metabolites, such as dihydrodiol epoxides that bind to and disrupt DNA and RNA, can induce possible resultant tumor formation [78].

The mean values of BaP$_{TEQ}$ (6.94 ± 4.36 mg/kg) and BaP$_{MEQ}$ (8.18 ± 4.05 mg/kg) of this study are higher than those reported in roadside dust in Nigeria ($3 \times 10^{-5}$–0.22 mg/kg; $5.2 \times 10^{-4}$–0.18 mg/kg [25]), the Buffalo Estuary in South Africa (1.213 mg/kg (mean) and 0.932 mg/kg (mean) [14]) and the Damodar river basin sediments in India (0.0019 mg/kg (mean) and 0.0016 mg/kg (mean) [4]). The total mean toxic equivalency quotient ($\sum$TEQ) (15.12 ± 8.4 mg/kg) of this study is also greater than the "safe level of 0.6 mg/Kg based on the risk-based soil criterion for the protection of human health from Canadian Council of Ministries of Environment" [79]; urban soil of Xi'an in Northwest China (0.423 mg/kg [9]);

Dhanbad in India (0.72 mg/kg [80]); and Lisbon in Portugal (0.229 mg/kg [81]). These comparisons show that PAHs in the sediments of the study area have high toxicity potency. In order to solve this problem, the appropriate regulatory bodies in the region must put to an end to the activities of these artisans at the site and carry out appropriate clean-up of the area.

Incremental lifetime cancer risk is an integrated approach utilized to assess carcinogenic PAHs by ingestion, dermal and inhalation means [4,14,18,25]. Each of the means of PAHs exposure was calculated for both children and adults, and a brief overview of the results is shown in Table 7. Cancer risks for children ranged from $2.85 \times 10^{-5}$ to $2.24 \times 10^{-4}$ (ingestion), $8.11 \times 10^{-5}$ to $7.20 \times 10^{-4}$ (dermal) and $4.00 \times 10^{-14}$ to $1.70 \times 10^{-13}$ (inhalation), while that of the adults ranged from $1.4 \times 10^{-6}$ to $8.10 \times 10^{-6}$ (ingestion), $3.31 \times 10^{-5}$ to $1.24 \times 10^{-4}$ (dermal) and $2.00 \times 10^{-13}$ to $7.0 \times 10^{-13}$ (inhalation). In all results, the dermal ILCR levels were higher than others (ILCRDerm > ILCRing > ILCRInh), and it is also higher for children than compared to adults. This finding is in line with those of [4,14,18,25]; thus, children are more prone to incremental lifetime cancer risk than adults as all other exposure media are also higher. Some of the reasons adduced to this include that, unlike adults, children are easily involved in the habit of extending their hands to their mouths; as such, the intake of PAH by children is believed to be higher than that by an adult, and the effects of high PAH in children are not made manifest due to their lower body weights, unlike the adults [9]. Again, as reported in similar studies [9,18,25,77,82,83], exposure of PAHs through inhalation is by far lower than the other means of exposure for this study.

**Table 7.** Cancer risk of PAHS in the river bed sediments of the study area.

| | **Adult** | | | |
| | ILCRing | ILCRDerm | ILCRInh | Total Cancer Risk |
| --- | --- | --- | --- | --- |
| ∑PAHs7c | $1.73 \times 10^{-4}$ | $2.29 \times 10^{-3}$ | $1.30 \times 10^{-11}$ | $2.46 \times 10^{-3}$ |
| Min | $1.40 \times 10^{-6}$ | $3.31 \times 10^{-5}$ | $2.00 \times 10^{-13}$ | $3.25 \times 10^{-5}$ |
| Max | $8.10 \times 10^{-6}$ | $1.24 \times 10^{-4}$ | $7.00 \times 10^{-13}$ | $1.32 \times 10^{-4}$ |
| Mean | $4.30 \times 10^{-6}$ | $5.72 \times 10^{-5}$ | $3.00 \times 10^{-13}$ | $6.15 \times 10^{-5}$ |
| | **Child** | | | |
| | ILCRing | ILCRDerm | ILCRInh | Total cancer risk |
| ∑PAHs7c | $2.67 \times 10^{-3}$ | $7.25 \times 10^{-3}$ | $3.18 \times 10^{-12}$ | $9.93 \times 10^{-3}$ |
| Min | $2.85 \times 10^{-5}$ | $8.11 \times 10^{-5}$ | $4.00 \times 10^{-14}$ | $1.10 \times 10^{-4}$ |
| Max | $2.24 \times 10^{-4}$ | $7.20 \times 10^{-4}$ | $1.70 \times 10^{-13}$ | $9.44 \times 10^{-4}$ |
| Mean | $6.70 \times 10^{-5}$ | $1.81 \times 10^{-4}$ | $8.00 \times 10^{-14}$ | $2.48 \times 10^{-4}$ |

By the regulatory guidelines [40], ILCR $\leq 10^{-6}$ values show no risk or negligible risk that is similar to that obtained from exposure to X-ray. While $\geq 10^{-4}$ values show high risk with adverse health effects such as cancer, the total ILCR value for each category is higher than $10^{-4}$. However, each of the means falls between $\leq 10^{-6}$ and ILCR $\geq 10^{-4}$; hence, it can be observed that the risk is above the baseline limits. Thus, there is a potential risk of cancer to the inhabitants of the area [4,9,14,24,25]. As earlier discussed, in solving this problem, the appropriate regulatory bodies and other organs of the government in the region are responsible for enforcing the outright stoppage of the activities of these artisans at the site and their responsibility to carry out appropriate clean-up of the area.

## 4. Conclusions

Polycyclic aromatic hydrocarbons (PAHs) are ubiquitous environmental pollutants that possess serious risks to human health and the environment. Forty riverbed sediments samples were collected in mangrove river bed sediments where artisanal refining of crude oil takes place in the Niger Delta of Nigeria. The samples were analysed for the concentra-

tion, occurrence, distribution, toxicity and health risk of sixteen priority PAHs (16PAHs) from the US EPA priority list. The result indicated that apart from Nap, Acy, BkF, InP and DbE, all the other PAHs were present in all the sampled points of the studied area, with BbF and BaA recording the highest mean values. The range and mean of the total PAHs ($\sum$16PAHs) of this study are 23.461–89.886 mg/kg and 42.607 $\pm$ 14.30 mg/kg dry weight, which is classified as heavily contaminated when compared to the European classification of PAHs pollution in soil. Again, the results are within the range of values reported in similar studies in the Niger Delta, but it is above those reported around the world in this study. The occurrence of the high molecular weight (HMW) PAHs (73.4%) supersedes those of the lower molecular weight (LMW) PAHs (26.6%), which also provides a clue about the source of the PAHs and the contamination status. Furthermore, the concentrations of the $\sum$16PAHs fall within the contaminated category when compared with the effect range factors as all the sixteen PAHs are higher than the effect range-low (ER-Low) values and more than 90% values of the 16PAHs are above ER-median (ER-M) values. The mean of the sixteen PAHs ($\sum$16PAHs) is higher than the ER-low (4.00 mg/kg), but it is slightly lower than the ER-median (44.792 mg/kg). These high values indicate potential risks to the environment and health of the citizens that subsist on the aquatic resources of the area.

The diagnostic ratios and principal component analysis suggest that the main contributors of PAHS into the sediments include the combustion of biomass, fossil fuel (crude oil) and pyrogenic sources. By using Benzo(a)pyrene toxic equivalency quotient (BaP$_{TEQ}$) (which is a test of carcinogenicity) and the mutagenic equivalency quotient (BaP$_{MEQ}$) (a mutagenicity test), human health risks due to exposure to PAHs of the study area were assessed. The result showed that mean BaP$_{TEC}$ (6.94 $\pm$ 4.36 mg/kg) and BaP$_{MEQ}$ (8.18 $\pm$ 4.05 mg/kg) values are significantly higher than some values reported by other researchers around the world. The order of significant contributors to BaP$_{TEC}$ values is BaP > DahA > BbF > BaA, while that of BaP$_{MEQ}$ is BaP > BbF > IndP > BaA. The total mean toxic equivalency quotient ($\sum$TEQ) (15.12 $\pm$ 8.4 mg/kg) is also greater than the safe level of 0.6 mg/kg based on the risk-based soil criterion for the protection of human health from the Canadian Council of Ministries of Environment. The result of incremental lifetime cancer risk (ILCR) showed that both adults and children can be affected more by dermal exposure than by ingestion and inhalation; moreover, children are more likely to be affected than the adults, as shown by the calculated Total$_{ILCR \text{ children}}$ ($2.48 \times 10^{-4}$) > Total$_{ILCR \text{ adults}}$ ($6.15 \times 10^{-5}$). Each of the means of exposure (adult and children) fall between ILCR $10^{-6}$ and $\geq 10^{-4}$; hence, the risk is above the baseline limit (ILCR $\leq 10^{-6}$); thus, there is a potential risk of cancer to the inhabitants of the area. In order to solve this problem, the first step is for the appropriate regulatory bodies and other organs of government in the region to enforce outright stoppage of the activities of these illegal artisans at the site and to carry out appropriate clean-up of the area.

**Author Contributions:** Conceptualization, I.C.H.; filed and laboratory analysis—I.C.H.; other field assistance and write up—I.C.H. and K.E.O.; review and editing—K.E.O. and K.T.; funding acquisition—K.T. All authors have read and agreed to the published version of the manuscript.

**Funding:** This research was supported by Prince of Songkla University and the Ministry of Higher Education, Science, Research and Innovation, Thailand, under the Reinventing University Project (Grant Number REV64007).

**Institutional Review Board Statement:** Not applicable.

**Informed Consent Statement:** Not applicable.

**Acknowledgments:** We (authors) wish to appreciate the efforts of Israel O. Clinton and all other field assistants that carried out fieldwork and also the staff and management of Ecotech Global Resources Ltd., Port Harcourt, and other Sister laboratories where the analysis was carried out.

**Conflicts of Interest:** The authors declare no conflict of interest. The funders had no role in the design of the study; in the collection, analyses or interpretation of data; in the writing of the manuscript; or in the decision to publish the results.

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
