# Peer review of "Toxicity and Risks Assessment of Polycyclic Aromatic Hydrocarbons in River Bed Sediments of an Artisanal Crude Oil Refining Area in the Niger Delta, Nigeria"

_water, doi:10.3390/w13223295_

Round 1

Reviewer 1 Report

This article reports a survey for analyzing polycyclic aromatic hydrocarbons in a river near an oil refining area in Nigeria. Concrete results have been shown based on a sampling campaign. The information from the results is meaningful. However, some major concerns are found. Please see my comment below.

General comments:

Double check the writing. There are many typos.

The authors should show background information to allow the audiences to understand the article easily. Such information includes but does not limit to principal component analysis, ER-L, ER-M, etc.

The article describes the results but with no in-depth discussion based on the results. The discussion should include (1) the level of contamination that is safe for human health and the environment, (2) the adverse effects of the contamination in the current situation, (3) how the problem can be solved with practical approaches. The discussion should be detailed.

Specific comments:

Lines 18 to 19: Explain what is sediment quality guideline effects range medium. Why does it indicate damages?

Lines 23 to 24: Explain the unit: mgTEQ/kg dw and mgMEQ/kg dw.

Lines 25 to 26: Show data for ILCRs.

Lines 69 to 71, “it is only the pyrolytic (Pyrogenic) and …”: This should be supported by quantitative information.

Lines 75 to 77: The adverse effect depends on the concentration. Pleas show the maximum level allowed in the environment.

Lines 90 to 93: You have discussed the adverse effects of PAHs previously. Please re-organize these paragraphs and combine the contents discussing the same topic together.

Line 99, “1821.5mgkg”: Make sure the unit is correct.

Line 100, “7681.39g/g dry weights (dw)”: Is the unit correct? It is weird.

Lines 115 to 123: This part to explain the background information of the studied area should be expanded. Please specify how much crude oil is discharged and how the ecological system degraded because of discharge. What is the maximum PAH level that is safe for the environment? You need to articulate the maximum level that is allowed.

Line 132: Please double check the pH at 4. Why is it so low?

Figure 1: Increase the resolution. It is blurred.

Section 2.2: It is better to take samples in different seasons. Did you take samples only once?

Line 143: It seems that 16 C in a fridge is high. Did any other studies use a so high temperature to store samples?

Section 2.7: Please provide more background information about ILCR. Based on Eqs (4) to (6), ILCR is a value depending on the contaminant concentrations that humans are exposed to. How can you distinguish the calculated ICLR is high or not?

Line 229: After you calculated three ILCRs, did you need to add them together? Please specify this and then indicate the criteria that determine the summed ILCR is high or low.

Line 239: What is oral slope factor?

Lines 272 to 273: Indicate here how molecular weight compounds are defined to be high and low. Does molecular weight connect to toxicity?

Lines 277 to 279: How did you determine these thresholds? Please specify the reason.

Line 281: I do not see COMBPAHs and the definition in Table 2.

Line 293: Please make sure the unit is correct.

Table 2: Please explain the short names, for example, ER-L and ER-M.

Lines 315 to 316: Please make this statement clear. Do you mean that HMW PAHs can be adsorbed by sediment more easily than LMW PAHs?

Figure 3: Explain how to read Fig 3(a). If both plots show a similar and simple data set, you do not need to have them. You can them remove them and use text in the article to describe the results.

Lines 334 to 339: I do not understand why these ratios can define the sources. Please discuss the mechanisms. If such discussion is background information, it can be made in Section 2.

Lines 348 to 350: Explain how this 5% was determined from BaA/(BaA + Chry)

Section 3.4: Please show background information about the principle of this analysis.

Lines 359 to 360: Explain how these three components are defined and identified.

Lines 361 to 363: I do not understand the logic. What are eight vectors and final coefficients?

Line 368: What is PAH Na? It is better to use your own words to describe the conclusion, rather than quoting the original sentence.

Lines 371 to 375, “Applying this information to our study…”: Rewrite this sentence to make it easier to be understood.

Lines 380 to 384, “The study area which is noted for…”: Rewrite this sentence to make it easier to be understood.

Table 5: You did not explain how these values were determined. How to calculate PC-1, PC-2 and PC-3? What do the negative values mean?

Figure 5: Explain how to read the figure. Does this figure show the same data set as Table 5? If yes, I do not think you need this figure.

Lines 394 to 395: Were the ER-L and ER-M values calculated based on your experimental data or obtained from somewhere? Please explain what ER-L and ER-M mean and why you used them.

Lines 400 to 401: Specify the future research and uncertainty to be considered.

Lines 411 to 413, “However, for both methods of estimation…”: Improve the writing of this sentence.

Lines 444 to 446, “The total BaPTEC and BaPMEQ values are significantly higher than…”: I do not see you mentioned this previously. Please add this conclusion to the previous text that discusses these values.

Reviewer 2 Report

The topic of the manuscript is within the scope of the journal and addresses an interesting topic.  What I could suggest for improving its significance is trying to extract inferences of general interest from your interpretations, which at the moment are limited to the regional scale. English grammar and syntax should be improved. More work should be done on presenting and discussing the results by giving a more international flavour. There is no detailed description of the lithological units and geochemical composition of rocks and sediments outcropping in the study area. This will help the audience to understand, among others, the fate and transport of the PAHs. Before applying the PCA, the authors have to report the results of measure of sampling adequacy, as well as the Bartlett's test of sphericity. The authors have to revise the PCA subsection by including a more detailed description of the calculation of the components.

Additional comments are inserted in the annotated .pdf file.

I hope these comments help to improve the quality of the manuscript.

Regards,

Round 2

Reviewer 1 Report

I appreciate the authors’ revision and responses. Some comments have not been addressed fully. Please see my follow-up comments below. General comments: My general comments are not mentioned in the Response to Reviewer’s Comment. Please specify how you improve the manuscript according to them. Response to Lines 18 to 19: In the abstract, please explicitly indicate what ER-M and ER-L mean to the ecosystem and public health. You showed ER-M and ER-L, but you did not show what they stand for and why they are important. The audiences will be confused. In Lines 243 to 249, please revise the expression to clearly indicate the levels for ER-M and ER-L. Response to Lines 25 to 26: Please convert scientific values to the normal format, i.e., 2.48 * 10^–4 Response to Lines 69 to 71: I see what you added. Please provide quantitative numbers from the cited references to demonstrate this conclusion. I think these papers have reported data to support this conclusion. Please provide their data. Response to Line 132: Do you know why pH can be as low as 4 for freshly deposited soft silt low tide? Response to Lines 277 to 279: You mentioned to see Line 297. But I do not see this in Line 297. Please make sure the correct citation is shown. Response to Lines 359 to 360: You need to indicate how PC1, PC2, and PC3 were defined and identified in the article. When I read the article, I feel confused about them. Response to Lines 371 to 375: I do not see you improved the sentence. The sentence is confusing. You need to improve the writing. Response to Table 5: Please explain what the negative values in Table 5 mean. Response to Figure 5: If you like to keep the figure, you need to discuss it and explain how it validates the data in Table 5.

Reviewer 2 Report

The manuscript is improved in relation to its initial version. There are still some points that require corrections. The abstract should be written in a way to attract an international audience. Please revise the abstract accordingly. Authors should remove blocks in italics or in quotation marks (" ") and revise accordingly. Additional comments are presented in the attached .pdf file.

Regards,
